# IL-25–induced shifts in macrophage polarization promote development of beige fat and improve metabolic homeostasis in mice

**Lingyi Li**[1,2☯], **Lei Ma**[1,2☯], **Zewei Zhao**[1,2☯], **Shiya Luo**[2], **Baoyong Gong**[3], **Jin Li**[4], **Juan Feng**[5], **Hui Zhang**[6], **Weiwei Qi**[2], **Ti Zhou**[2], **Xia Yang**[2], **Guoquan Gao**[2], **Zhonghan Yang**[1,2]*

1 Department of Biochemistry, Molecular Cancer Research Center, School of Medicine, Sun Yat-sen University, Shenzhen, Guangdong Province, China, 2 Department of Biochemistry, Zhongshan School of Medicine, Sun Yat-Sen University, Guangzhou, Guangdong Province, China, 3 Guangdong Provincial Key Laboratory of Laboratory Animals, Guangdong Laboratory Animals Monitoring Institute, Guangzhou, Guangdong Province, China, 4 Department of Gerontology, the First Affiliated Hospital, Sun Yat-Sen University, Guangzhou, Guangdong Province, China, 5 School of Stomatology, Foshan University, Foshan, Guangdong Province, China, 6 Metabolic Innovation Center, Sun Yat-sen University, Guangzhou, Guangdong Province, China

☯ These authors contributed equally to this work.
* yangzhh@mail.sysu.edu.cn

**Data Availability Statement:** All relevant data are within the paper and its Supporting Information files.

## Abstract

Beige fat dissipates energy and functions as a defense against cold and obesity, but the mechanism for its development is unclear. We found that interleukin (IL)-25 signaling through its cognate receptor, IL-17 receptor B (IL-17RB), increased in adipose tissue after cold exposure and β3-adrenoceptor agonist stimulation. IL-25 induced beige fat formation in white adipose tissue (WAT) by releasing IL-4 and IL-13 and promoting alternative activation of macrophages that regulate innervation and up-regulate tyrosine hydroxylase (TH) up-regulation to produce more catecholamine including norepinephrine (NE). Blockade of IL-4Rα or depletion of macrophages with clodronate-loaded liposomes in vivo significantly impaired the beige fat formation in WAT. Mice fed with a high-fat diet (HFD) were protected from obesity and related metabolic disorders when given IL-25 through a process that involved the uncoupling protein 1 (UCP1)-mediated thermogenesis. In conclusion, the activation of IL-25 signaling in WAT may have therapeutic potential for controlling obesity and its associated metabolic disorders.

## Introduction

Obesity, which affects approximately 13% of adults worldwide, has become a major and pressing global health problem. Obesity increases the risk of various metabolic disorders such as hypertension, type 2 diabetes, and cardiovascular diseases as well as cancer [1,2]. Recent studies have demonstrated that adipose tissue contains a diverse cellular composition and that

**Funding:** This work was funded by the National Nature Science Foundation of China(http://www. nsfc.gov.cn/, grant number No: 81570764, 81770808, 81701414, 81872165 and 81871211), National Key R&D Program of China (http://www. nsfc.gov.cn/, grant number No. 2018YFA0800403), Guangdong Provincial Key R&D Program (http:// gdstc.gd.gov.cn/, grant number No: 2018B030337001 and 2019B020227003), Key Project of Nature Science Foundation of Guangdong Province, China (http://gdstc.gd.gov. cn/, grant number No. 2019B1515120077), Guangdong Natural Science Fund (http://gdstc.gd. gov.cn/, grant number No: 2019A1515011810 and 2020A1515010365), Guangdong Science and Technology Project (http://gdstc.gd.gov.cn/, grant number No. 2017A020215075), Guangdong Provincial Key Laboratory of Precision Medicine and Clinical Translation Research of Hakka Population (http://gdstc.gd.gov.cn/, grant number No. 2018B030322003KF01), Guangzhou Science and Technology Project (http://kjj.gz.gov.cn/, grant number No: 201807010069, 201803010017 and 202002020022) and Shenzhen Science and Technology Project (http://stic.sz.gov.cn/, grant number No. JCYJ20190807154205627) received by Weiwei Qi, Ti Zhou, Xia Yang, Guoquan Gao and Zhonghan Yang. The funders had no role in study design, data collection and analysis, decision to publish, or preparation of the manuscript.

**Competing interests:** The authors have declared that no competing interests exist.

**Abbreviations:** AP, adipocyte precursor; BAT, brown adipose tissue; CL, CL-316, 243; CLAMS, Comprehensive Lab Animal Monitoring System; DIO, diet-induced obesity; eWAT, epididymal WAT; FFA, free fatty acid; GTT, glucose tolerance test; HE, hematoxylin and eosin; HFD, high-fat diet; IL, interleukin; IL-17RB, IL-17 receptor B; ILC2, type 2 innate lymphoid cell; IP, intraperitoneally; ITT, insulin tolerance test; iWAT, inguinal WAT; Myf5, myogenic factor 5; NCD, normal chow diet; NE, norepinephrine; NF, neurofilament; Pax7, paired box 7; Pdgfr-α, platelet-derived growth factor receptor alpha; RT-qPCR, Reverse transcription - quantitative PCR; scWAT, subcutaneous WAT; SVF, stromal vascular fraction; TG, triglyceride; TH, tyrosine hydroxylase; UCP1, uncoupling protein 1; WAT, white adipose tissue; WT, wild-type.

metabolic pathways can be modulated by cytokine signaling networks that represent targets for the control of obesity. Two types of thermogenic adipocytes that control energy and glucose homeostasis in humans include classical brown adipocytes (developmental origin from Myf5$^+$ cells) and beige (also referred to as brite) adipocytes (origin from Myf5$^-$ cells), and both cell populations contain uncoupling protein 1 (UCP1) [3]. Activation of UCP1-positive adipocytes can release heat via the uncoupled oxidation respiratory chain for ATP synthesis [4]. The lineage of beige adipocytes that are composed of heterogeneous cell populations in multiple white adipose tissue (WAT) depots can exhibit UCP1-dependent thermogenic capacity when stimulated by certain external cues, such as chronic cold exposure, exercise, and long-term treatment with PPARγ agonist (e.g., rosiglitazone), while classical brown adipocytes in the interscapular region gradually disappear with age [3,5]. $^{18}$F-FDG-PET/CT scans have identified active brown adipose tissue (BAT) depots in the cervical, supraclavicular, axillary, and paravertebral regions of adult humans. Thus, the development and metabolic regulation of these thermogenic adipocyte populations in fat tissue could also be a potential target for the treatment of obesity [6,7].

Although both of these adipocyte populations are similar in high expression of UCP1, classical brown adipocyte are derived from a myogenic factor 5 (Myf5) and paired box 7 (Pax7)-positive precursors [8], while the precursors of beige adipocytes are from a Myf5-negative, platelet-derived growth factor receptor alpha (Pdgfr-α)-positive cell lineage [9]. In addition, beige adipocytes with the browning genes *Ucp1*, *Ppargc1a*, and *CD81* are activated by cold and β-agonist stimulation and located mainly in the subcutaneous and epididymal fat depots [4,10–12], while brown adipocytes are primarily located in the neck and interscapular region and express high levels of UCP1. Further, the development of brown adipocytes and beige adipocytes was shown to be regulated by different pathways [10,13]. Notably, the beige adipose tissue innervation depot shares a coincident shift in the gene expression profile of neurons in stellate ganglion projecting to the BAT depot [14].

Adipose tissue is an endocrine and immune organ including different kinds of immune cells such as macrophages, eosinophils, T cells, and B cells [15] that contribute to the metabolic condition of the whole organism through effects on adipose cell metabolism and immune regulation. Crosstalk among adipocytes, recruited immune cells and released cytokines play a key role in maintaining metabolic homeostasis. Emerging evidence shows that the immune environment also influences the modulation of adipose tissue, especially in the biogenesis of beige adipocytes. For example, eosinophils, γδ T cell, and alternatively activated macrophages in adipose tissue are crucial for the cold exposure–induced browning of WAT [16–18]. In addition, mast cells may also influence the development of beige fat by sensing cold environment and subsequently releasing histamine to promote UCP1 expression [19,20]. Therefore, although the underlying mechanisms are not fully understood, the diversity of endocrines and cytokines may play a key role in the modulation of adipose tissue, and immune cells may stimulate the biogenesis of beige fat [21]. Previously, we found that nematode infection stimulated type 2 immunity by releasing macrophage-responsive Th2 cytokines including interleukin (IL)-25 [22] that modulated body weight and metabolic dysfunction associated with obesity [23]. To further explore whether this process above is related to the thermogenesis of adipocyte populations, we studied the role of IL-25 in modulating the browning effect of adipose tissue.

IL-25 is a member of IL-17 cytokine family (also called IL-17E) present in various tissues. The receptor of IL-25, IL-17 receptor B (IL-17RB), is expressed in various cell types, such as epithelial cells, eosinophils, and NKT cells [24]. In addition to its role in mucosal and type 2 immunity, IL-25 contributes to protection against high-fat diet (HFD)-induced hepatic steatosis [25], excessive lipid accumulation in the liver, and regulation of lipid metabolism to modulate the body weight via alternatively activate macrophages [26]. Whether IL-25 is also

involved in the biogenesis of brown-in-white (brite)/beige adipocytes and associated metabolic disorders is not known.

Given that the brown and beige adipose tissue can increase thermogenesis by uncoupling oxidative phosphorylation through UCP1 up-regulation, we hypothesize that constitutively expressed IL-25 induces beige adipose tissue formation and improves metabolic homeostasis and thermogenesis and that enhanced IL-25 production would protect against insulin resistance. The present study was aimed at investigating (1) the effects of β-agonist or cold exposure on IL-25 expression in the adipose tissue; (2) the effects of IL-25 on brite/beige adipocytes in WAT; (3) the role of macrophages in this process that IL-25 induces the beige fat; and (4) whether IL-25 restores the homeostasis against insulin resistance.

## Results

### IL-25 signaling increased in beige adipose tissue upon β3-adrenoceptor agonist stimulation or cold exposure

Administration of the β3-adrenoceptor agonist (CL-316, 243 (CL)) to 8-week-old wild-type (WT) C57BL/6J mice induced a robust browning phenotype characterized by high expression of UCP-1 and development of multilocular adipocytes in the adipose tissue was found (Fig 1A, S1A–S1C Fig). Expression of UCP1 in subcutaneous WAT (scWAT) was much higher than that of epididymal WAT (eWAT) (S1A Fig). Time course analysis showed that CL increased expression of IL-25 and its cognate receptor, IL-17RB, in scWAT (Fig 1B and 1C) and eWAT (S1D and S1E Fig). The CL-induced expression of IL-25 was further confirmed in scWAT (Fig 1G, left panel) and eWAT (S1F Fig, left panel) by ELISA. Fig 1H shows that higher expression of IL-25 in adipose tissue does not increase circulatory IL-25.

Next, WT mice were separated into 2 groups and housed separately at 22˚C or 4˚C, respectively, for 2 days. Cold exposure induced a comparable beige phenotype in scWAT characterized by up-regulation of UCP1 (Fig 1D). However, the elevation of UCP1 was not observed in eWAT (S1G–S1I Fig), suggesting that cold-induced sympathetic activation preferentially stimulated the browning in scWAT only. Similarly, the expression of IL-25 and IL-17RB was examined at different temperature. Concomitant to the browning of WAT, cold exposure preferentially resulted in an increase of IL-17RB (Fig 1E and 1F) and IL-25 (Fig 1G, right panel) in scWAT. As expected, cold exposure did not increase circulatory IL-25 (Fig 1H, right panel). We examined the sources of IL-25 and IL-17RB in adipose tissue and found that IL-25 was secreted mainly by adipocytes and less by macrophages (Fig 1I, 1K and 1L), while IL-17RB was mainly expressed on macrophages, but less in adipocytes (Fig 1J and 1L). Unexpectedly, cold exposure only increased expression of IL-25 and IL-17RB in scWAT but not in eWAT (S1F and S1J–S1L Fig) and decreased expression in the liver (S1L Fig).

### Administration of IL-25 induced browning of scWAT and repressed chronic low-grade inflammation in mice with diet-induced obesity (DIO)

To further investigate whether IL-25 could induce thermogenic gene expression and browning of WAT, recombinant IL-25 was intraperitoneally (IP) injected into mice fed with a normal chow diet (NCD) or with an HFD for 7 days. Fig 2A shows that IL-25 increased the expression of UCP1 protein in scWAT but not in BAT in a dose-dependent manner. As the expression of UCP1 in eWAT also increased after treatment of mice with a high dose of IL-25 (1,000 ng/d) (Fig 2A), this dose was used in the following experiment. In addition, the IL-25–induced up-regulation of UCP1 gene expression was similar to that of IL-4 (1,000 ng/d), a positive control to induce the browning of adipose tissue (S3A Fig). Reverse transcription - quantitative PCR (RT-qPCR)

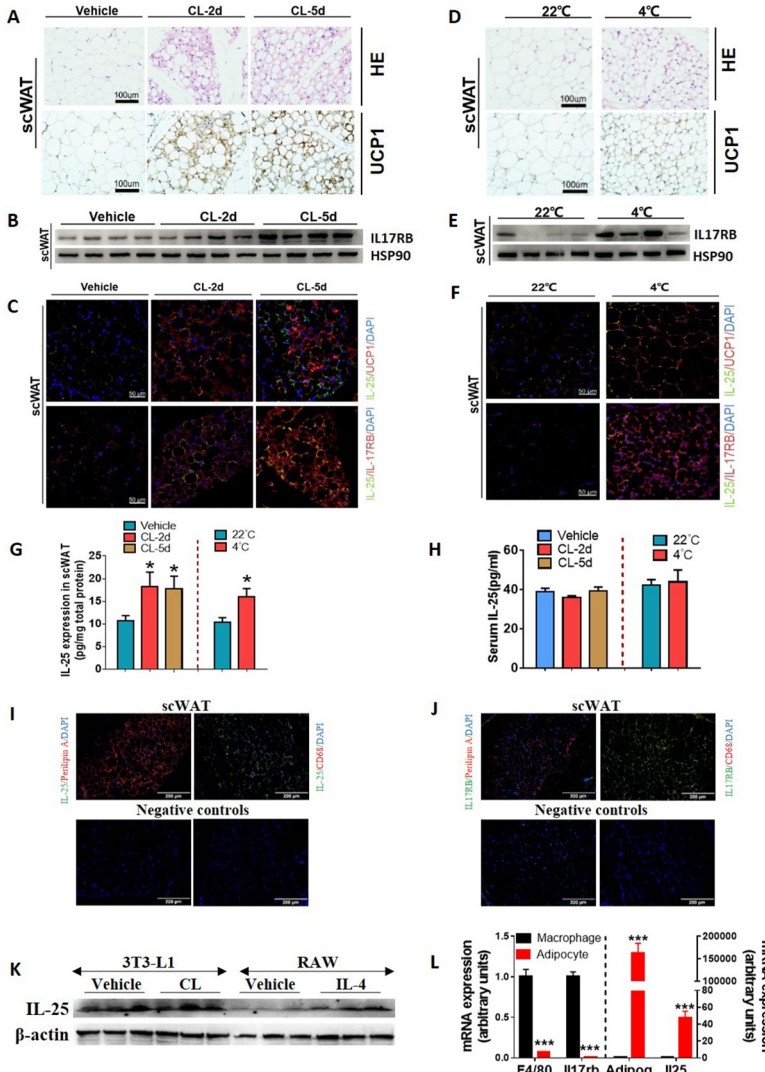

**Fig 1. IL-25 signaling increased in subcutaneous beige adipose tissue induced by β3-adrenergic agonist stimulation or cold exposure.** C57BL/6J WT mice were injected with CL (1 mg/kg body weight) for 2 (CL-2d) and 5 days (CL-5d) or were placed at controlled temperature of either 22°C or 4°C (cold challenge) in individual cages for 48 hours (*n* = 4–5 per treatment). (A, D) H&E staining and immunohistochemical staining for UCP1 in scWAT of mice treated with CL for 5 days or placed at controlled temperatures for 48 hours (*n* = 4–5 per treatment). 400× magnification. (B, E) The protein level of IL-17RB and HSP90 was analyzed by western blot in scWAT of mice treated with CL for 5 days or placed at controlled temperature (22°C) and cold challenge (4°C) in individual cages for 48 hours (*n* = 4–5 per treatment). HSP90 was used a loading control. (C, F) Immunofluorescent staining for IL-25 (IL-25+ green) and UCP1 (UCP1+ red) or IL-17RB (IL-17RB+ red) in scWAT in mice treated by CL for 5 days or placed at controlled temperatures for 48 hours (*n* = 4–5 per treatment). Nucleus stained with DAPI (blue). Images were photographed at 200× magnification. (G, H) IL-25 protein expression in scWAT or serum (*n* = 4–5 per treatment) in mice treated by CL for 2 days, 5 days, or placed at controlled temperature (22°C) and cold challenge (4°C) in individual cages for 48 hours (*n* = 4–5 per treatment). (I, J) Immunofluorescent staining for IL-25 (IL-25+ green) or IL-17RB (IL-17RB+ green) with CD68 (CD68+ red) or Perilipin A (Perilipin A+ red) of scWAT in mice placed at room temperature (22°C) (*n* = 4–5 per treatment). Nucleus stained with DAPI (blue). Images were photographed at 200× magnification. (K) The protein level of IL-25 and β-actin was analyzed by western blot in 3T3-L1 cells treated with CL or PBS for 2 days or in RAW cells treated with IL-4 or vehicle for 2 days. (L) qPCR analysis of F4/80, Adiponectin, IL-17RB, and IL-25 in primary adipocytes and macrophages. The data underlying this figure can be found in S1 Data. CL, CL-316, 243; H&E, hematoxylin and eosin; IL, interleukin; IL-17RB, IL-17 receptor B; qPCR, quantitative PCR; scWAT, subcutaneous WAT; UCP1, uncoupling protein 1; WAT, white adipose tissue; WT, wild-type.

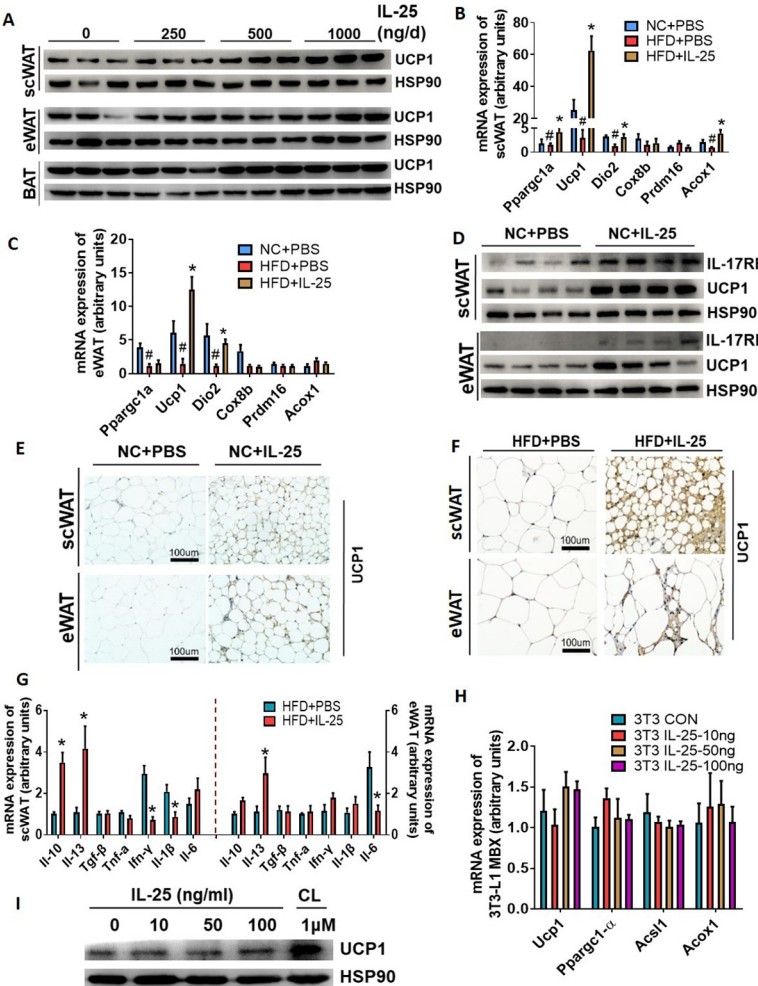

**Fig 2. IL-25 induced the browning of WAT and affected inflammatory cytokines in vivo independent of a direct action on adipocytes.** (A) Immunoblot was used to quantify the expression of UCP1 (3 representative bands are shown) in adipose tissues of WT mice administrated with various doses of IL-25 for 7 days ($n = 5$) (The auto exposure time of eWAT was significantly longer than scWAT). (B–G) WT mice fed with NCD or HFD for 12 weeks were injected with IL-25 (1 μg/day) over 7 days. (B, C) qPCR analysis of markers associated with β-oxidation and thermogenesis in scWAT (B) and eWAT (C). (D) Immunoblot analysis of IL-17RB and UCP1 protein in WAT. (E, F) Representative images of WAT stained for UCP1 from mice fed with NCD (E) or with HFD (F). 200× magnification (top), 400× magnification (bottom). (G) C57/BL6J mice ($n = 5$) fed with an HFD for 12 weeks were administrated with vehicle or IL-25 (1 μg/day) over 14 days and then qPCR analysis of genes associated with pro/anti-inflammatory cytokines in scWAT and eWAT. (H, I) Differentiated 3T3-L1 MBX cells (adipocytes) were treated with various doses of IL-25 or CL. (H) The mRNA expression of thermogenesis or β-oxidation genes. (I) Immunoblot analysis of UCP1 protein in adipocyte. The data underlying this figure can be found in S1 Data. BAT, brown adipose tissue; CL, CL-316, 243; eWAT, epididymal WAT; HFD, high-fat diet; IL, interleukin; IL-17RB, IL-17 receptor B; NCD, normal chow diet; qPCR, quantitative PCR; scWAT, subcutaneous WAT; UCP1, uncoupling protein 1; WAT, white adipose tissue; WT, wild-type.

analysis showed that IL-25 induced the expression of beige fat–associated genes (Fig 2B and 2C). Furthermore, to validate the activity of recombinant IL-25, its receptor, IL-17RB, was also examined and shown to be up-regulated (Fig 2D). Histologic analysis showed that beige fat was characterized by high expression of the specific mitochondrial UCP1 and multilocular morphology in scWAT and eWAT after IL-25 treatment (Fig 2E and 2F). These data suggested that IL-25 stimulated the development of beige fat.

We next analyzed whether IL-25 suppressed the expression of pro-inflammatory genes and increased the expression of anti-inflammatory genes in the chronic inflammatory adipose tissue of mice with DIO. Fig 2G shows that IL-25 promoted the expression of anti-inflammatory cytokines IL-10 and IL-13 and decreased the expression of pro-inflammatory cytokines IFN-γ, IL-1β, and IL-6.

Next, to further explore the mechanism of IL-25–induced development of beige fat, recombinant IL-25 was added to cultures of differentiated 3T3-L1 MBX adipocytes in vitro, and CL was added to induce the expression of thermogenic or β-oxidation genes as a positive control. Fig 2H and 2I showed that IL-25 did not augment the expression of thermogenic or β-oxidation genes such as *Ucp1*, *Pgc-1α*, *Acsl1*, and *Acox1* at various doses of treatment in vitro.

## IL-25 increased the level of IL-4 to stimulate the browning of adipose tissue via inducing alternatively activated macrophages

Injection of WT mice with IL-25 induced macrophage phenotype switch from pro-inflammatory classically activated macrophages (iNOS as a marker) to anti-inflammatory alternatively activated macrophages (ARG1 and YM1 as markers) in WAT (Fig 3A). IL-25 also increased IL-4 and IL-13 (Fig 3B and 3C), which is important for induction of alternative activation programing of macrophages in adipose tissue. However, these changes were not found when WT mice were injected with irisin (Fig 3D), a myokine that induces the browning of adipose tissue by direct actions on adipocytes [27].

To explore the effects of IL-25 and IL-4 on the commitment of adipocyte precursors (APs) to develop the beige fat lineage, APs were isolated from scWAT, treated with IL-4 or IL-25 in vitro, and analyzed for markers of beige adipocytes. Fig 3E shows that APs treated with IL-4 expressed UCP1. In contrast, APs that were co-cultured with peritoneal macrophages treated with IL-25 showed enhanced UCP1 expression that was blocked in the presence of anti-IL-4Rα neutralizing antibody (Fig 3F). Next, anti-IL-4Rα antibody was IP injected into IL-25–treated mice fed with NCD. Disruption of IL-4/IL-13 signaling with anti-IL-4Rα neutralizing antibody in vivo decreased the IL-25–induced mRNA expression of Arg-1 and YM-1 genes that are indicative of alternatively activated macrophages in adipose tissues (Fig 3G). Western blot and RT-qPCR analysis of adipose tissues showed that the expression of UCP1 was increased in WT mice treated with IL-25, while injection of anti-IL-4Rα antibody blunted the expression of UCP1 (Fig 3G and 3H).

## IL-25 regulated adipose tissue innervation by macrophages

We next investigated the effect of IL-25 on the sympathetic nerve innervation in adipose tissue. The expression of tyrosine hydroxylase (TH) was induced in scWAT and eWAT after injection of IL-25 in WT mice in vivo (Fig 4A–4C) along with the induction of UCP1. Histological staining of TH and neurofilament (NF) in scWAT and eWAT showed more transverse section of sympathetic axons in the IL-25–treated mice than those in the PBS-treated mice (Fig 4D) or upon cold exposure (Fig 4F, left panel). Similarly, IL-25 also increased the level of norepinephrine (NE) in scWAT and eWAT (Fig 4E). These results showed that IL-25 significantly enhanced the innervation of scWAT and eWAT and induced the release of NE.

We further tested an IL-25–deficient (IL-25$^{-/-}$) mouse model and found that the effect of cold exposure on sympathetic distribution was blocked in IL-25$^{-/-}$ mice (Fig 4F, right panel). To further investigate whether BAT macrophages regulate tissue innervations, we injected clodronate-loaded liposomes into mice with DIO to deplete macrophages and found that depletion of macrophages blunted the IL-25–induced expression of TH and UCP1 (Fig 4G).

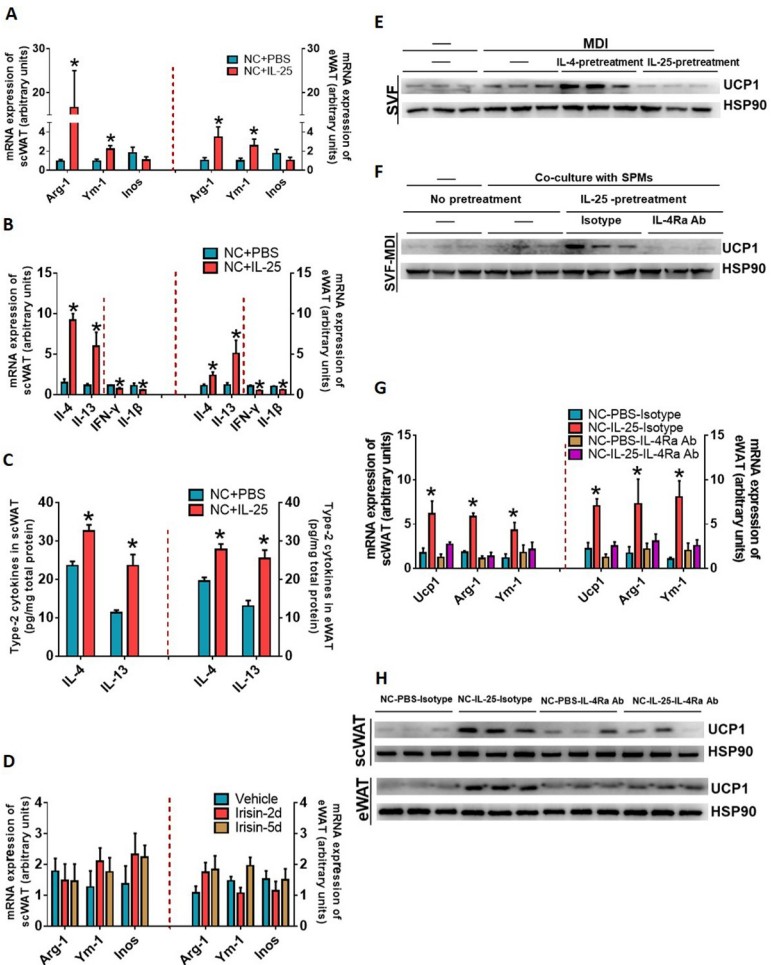

**Fig 3. IL-25 stimulated IL-4/IL-13 release to promote the browning of adipose tissue via inducing alternatively activated macrophages.** (A–C) C57/BL6J WT mice ($n = 5$) were injected with vehicle or IL-25 (1 µg/day) over 7 days and then (A) qPCR analysis of markers associated with macrophage polarization and (B) the mRNA expression of IL-4/IL-13 or IFN-γ/IL-1β genes in WAT. (C) IL-25 stimulates IL-4/IL-13 release in WAT. (D) C57/BL6J WT mice ($n = 4$–5) were injected with vehicle or irisin (1 µg/day) for 3 or 5 days, and genes associated with macrophage polarization were analyzed in WAT. (E) UCP1 examined by western blot in differentiated APs (MDI-induced SVF) pretreated with IL-4 (50 ng/ml) or IL-25 (50 ng/ml). (F) UCP1 examined by western blot in differentiated APs co-cultured with peritoneal macrophages stimulated with IL-25 (50 ng/ml) and anti-IL-4Rα neutralizing antibody (50 µg/ml). (G, H) C57/BL6J mice ($n = 5$) injected with vehicle or IL-25 (1 µg/day) for 7 days and anti-IL-4Rα neutralizing antibody (125 µg/day) or isotype control were injected at day 1 and day 4. Real-time PCR analysis of mRNA level of Ucp1, Arg-1, and Ym-1 (G) and western blot analysis of UCP1 protein level in WAT (H). HSP90 was used as a loading control. *$p < 0.05$ by two-sided unpaired $t$ test. Data present as mean ± SEM. The data underlying this figure can be found in S1 Data. AP, adipose precursor; eWAT, epididymal WAT; IL, interleukin; qPCR, quantitative PCR; scWAT, subcutaneous WAT; UCP1, uncoupling protein 1; WAT, white adipose tissue; WT, wild-type.

## IL-25 improved the metabolic homeostasis of mice with DIO

The effect of IL-25 treatment of mice with DIO on weight gain and related indexes of lipid and glucose metabolism was investigated. We found that mice fed with an HFD and treated with IL-25 gained significantly less weight (Fig 5A) compared with HFD-fed mice not treated with IL-25. The lipid droplet size and adipocyte size of scWAT and eWAT were also decreased in the IL-25–treated mice fed with the HFD (Fig 5B–5D). Furthermore, IL-25 decreased the blood glucose and circulating insulin in the mice fed with the HFD (Fig 5E), significantly

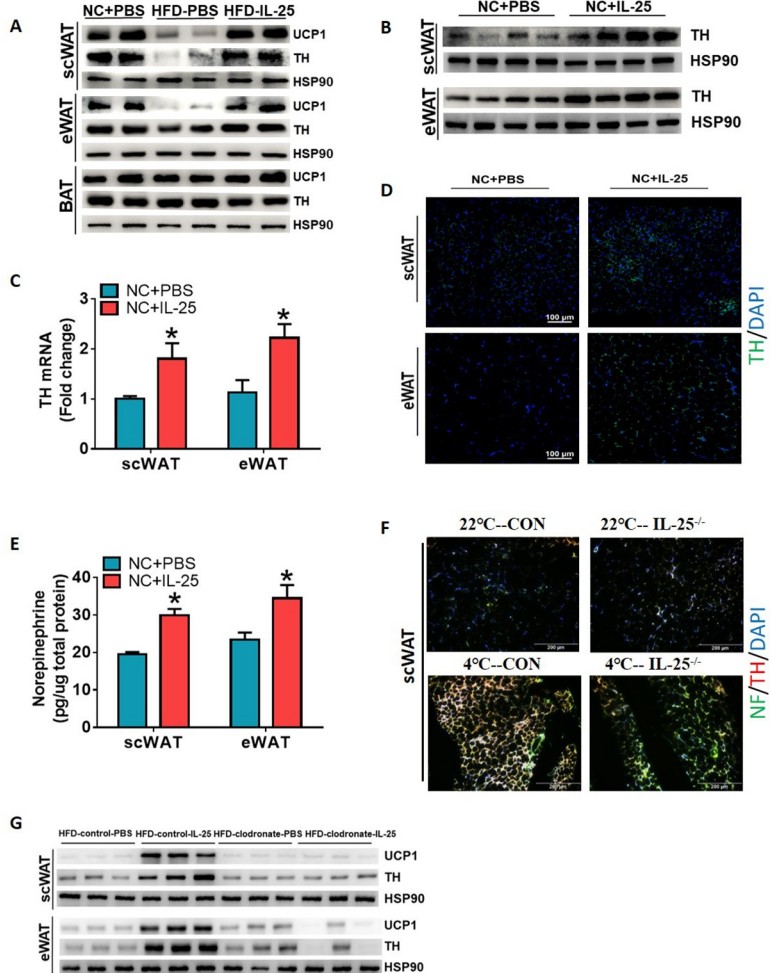

**Fig 4. IL-25 regulated the innervation of WAT by macrophages.** (A–D) C57/BL6J mice ($n$ = 5) or mice with DIO ($n$ = 5) injected with vehicle or IL-25 (1 μg/day) for 7 days. (A) Western blot analysis for level of TH and UCP1 protein in scWAT, eWAT, and BAT of mice fed with HFD. (B) Western blot analysis of TH protein level and (C) real-time PCR analysis of mRNA level of TH in WAT of mice fed with an NCD. (D) Representative immunofluorescence images of TH in WAT. (E) Total NE production in WAT measured by ELISA. (F) WT mice and IL-25$^{-/-}$ mice were placed at controlled temperature of 22˚C or 4˚C in individual cages for 5 days ($n$ = 4–5 per treatment). Representative immunofluorescence images of TH/NF in scWAT. (G) Mice with DIO ($n$ = 5) injected with clodronate-loaded liposomes to obliterate macrophages and then injected with vehicle or IL-25 (1 μg/day) for 14 days. Western blot analysis of TH and UCP1 protein in WAT. $^*p < 0.05$ by two-sided unpaired $t$ test. Data present as mean ± SEM. The data underlying this figure can be found in S1 Data. BAT, brown adipose tissue; DIO, diet-induced obesity; eWAT, epididymal WAT; HFD, high-fat diet; IL, interleukin; NCD, normal chow diet; NE, norepinephrine; NF, neurofilament; scWAT, subcutaneous WAT; TH, tyrosine hydroxylase; UCP1, uncoupling protein 1; WAT, white adipose tissue; WT, wild-type.

improved glucose disposal (Fig 5F) and insulin sensitivity (Fig 5G), and restored normal insu-lin stimulating p-AKT activity in the eWAT, liver, and muscle (Fig 5H).

## Depletion of IL-25 reduced the metabolism of glucose, lipid, and energy expenditure

IL-25$^{-/-}$ mice gained more weight (Fig 6A) than WT mice when fed with an HFD for 12 weeks, and the size of the lipid droplets in scWAT was also increased after cold stimulation at 4˚C for 48 hours (Fig 6B). We examined the relevant index of glucose metabolism and found

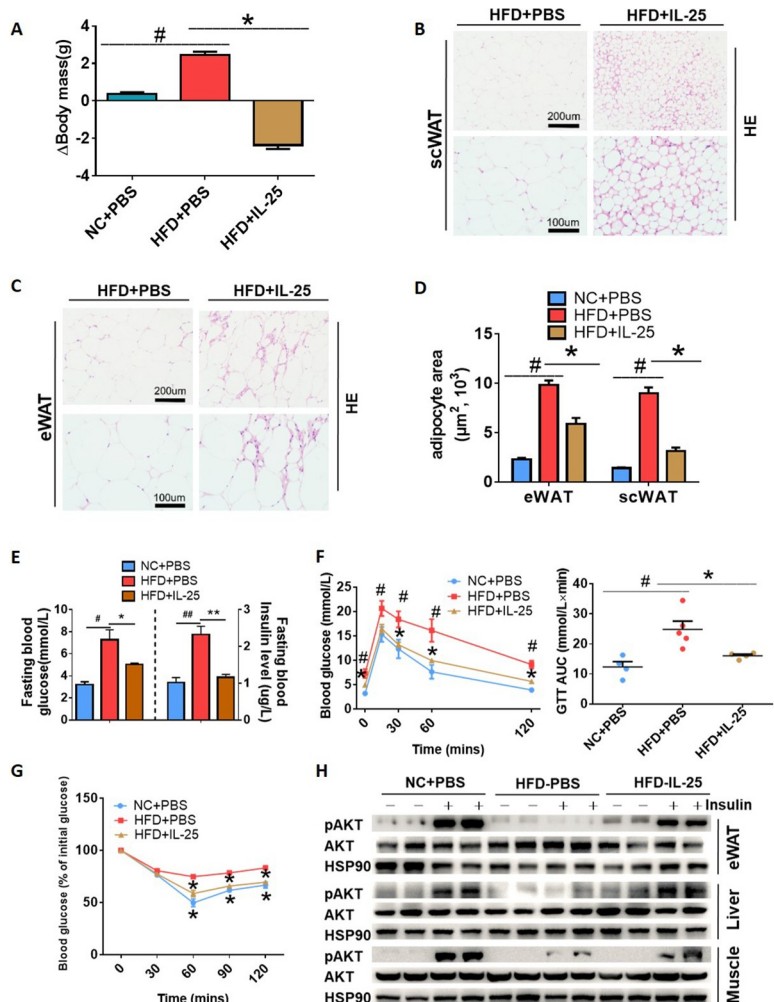

**Fig 5. IL-25 improved the metabolic homeostasis of mice with DIO against obesity and insulin resistance.** (A–H) C57/BL6J mice fed with an NCD or HFD for 12 weeks (*n* = 5 per treatment) were injected with vehicle or IL-25 over 14 days. (A) The change in body mass in mice with different treatments. (B, C) Representative images of scWAT (B) and eWAT (C) stained with H&E. 200× magnification (top), 400× magnification (bottom). (D) The adipocyte area of scWAT or eWAT in HFD-induced obese mice injected with vehicle or IL-25 for 14 days. (E) The fasting blood glucose and insulin in HFD-induced obese mice injected with vehicle or IL-25 for 14 days. (F) GTT was conducted by intraperitoneal injection of glucose (2 g·kg$^{-1}$) and measurement of blood glucose concentration with a OneTouch Ultra Glucometer at designed time points in overnight-fasted mice. (G) ITT was done by IP injection of insulin (0.75 U·kg$^{-1}$) and measurement of blood glucose concentration by a OneTouch Ultra Glucometer at designed time points in 8 h–fasted mice. (H) Western blot analysis of the phosphorylation of AKT and total AKT in eWAT, liver, and muscle. The tissues were harvested within 10 minutes after an injection of insulin (0.5 U·kg$^{-1}$). #$p < 0.05$, compared with NCD group, *$p < 0.05$, compared with HFD group by two-sided unpaired *t* test. Data present as mean ± SEM. The data underlying this figure can be found in S1 Data. auc, area under the curve; DIO, diet-induced obesity; eWAT, epididymal WAT; GTT, glucose tolerance test; HE, hematoxylin and eosin; HFD, high-fat diet; IL, interleukin; IP, intraperitoneal; ITT, insulin tolerance test; NCD, normal chow diet; scWAT, subcutaneous WAT.

that glucose disposal and insulin sensitivity were enhanced in IL-25$^{-/-}$ mice fed with an HFD compared with WT controls (Fig 6C–6E). Furthermore, oxygen consumption (Fig 6F), carbon dioxide production (Fig 6G), respiratory entropy (Fig 6H), and calorie consumption (Fig 6I), measured by Comprehensive Lab Animal Monitoring System (CLAMS), and energy expenditure were significantly reduced in IL-25$^{-/-}$ mice fed with an HFD compared with control. In addition, we also investigated the changes of metabolic homeostasis after cold exposure in IL-

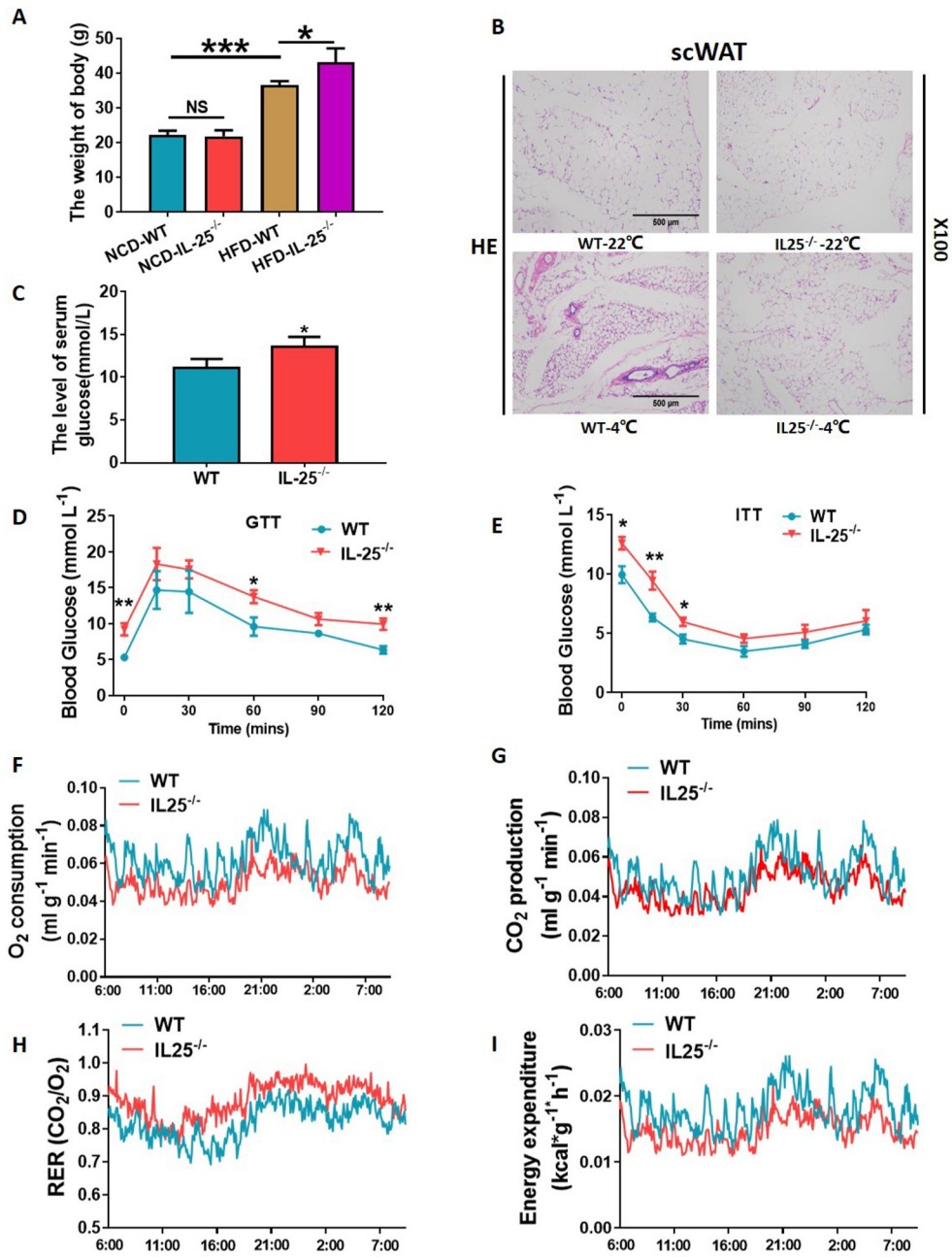

**Fig 6. Depletion of IL-25 altered the metabolism of glucose, lipid, and energy expenditure.** (A) The change of body weight in IL-25$^{-/-}$ mice and control mice that were fed with an NCD or HFD. (B) WT and IL-25$^{-/-}$ mice were maintained at 22°C or 4°C for 48 hours. The HE staining of IL-25$^{-/-}$ mice and control mice in scWAT. (C) The level of serum glucose in IL-25$^{-/-}$ mice and control mice. The GTT (D) and ITT (E) of the IL-25$^{-/-}$ mice and control mice by IP injection of glucose (1 g·kg$^{-1}$) or injection of insulin (1 U·kg$^{-1}$) and measurement of blood glucose concentration in overnight-fasted mice. The $O_2$ consumption (F), $CO_2$ production (G), respiratory quotient (H), and the calorie consumption (I) of IL-25$^{-/-}$ mice and control mice. $^*p < 0.05$, compared with control group by two-sided unpaired $t$ test. Data present as mean ± SEM. The data underlying this figure can be found in S1 Data. GTT, glucose tolerance test; HE, hematoxylin and eosin; HFD, high-fat diet; IL, interleukin; IP, intraperitoneal; ITT, insulin tolerance test; NCD, normal chow diet; NS, not significant; RER, respiratory exchange ratio; scWAT, subcutaneous WAT; WT, wild-type.

25$^{-/-}$ mice and found that IL-25 played an important role in the improvement of metabolic homeostasis during the cold exposure (S2 Fig).

## Macrophage depletion and UCP1 gene knockout (UCP1$^{-/-}$) impaired IL-25–mediated improvement in glucose homeostasis in mice

To investigate whether IL-25 regulates its anti-obesity and antidiabetic effects through macrophages, we injected mice with DIO with clodronate-loaded liposomes to deplete macrophages in adipose tissue (S3B Fig) and also injected IL-25. IL-25 did not reduce fasting blood glucose (Fig 7A) or improve glucose tolerance (Fig 7B) and insulin sensitivity (Fig 7C) in mice with

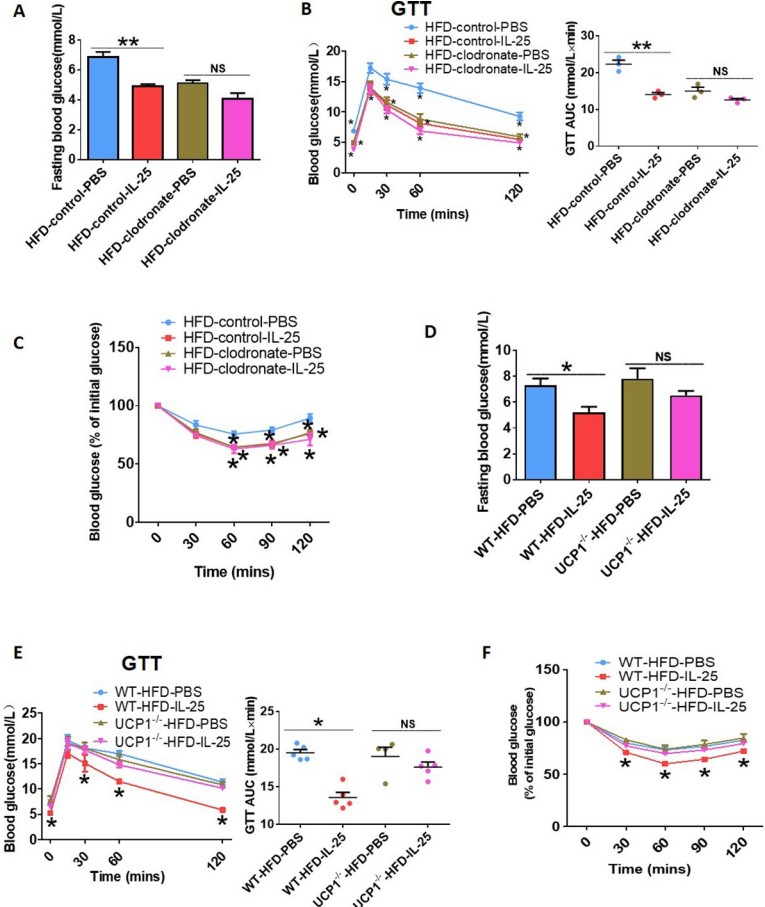

**Fig 7. Depletion of macrophages and genetic deletion of UCP1 impaired the improvement of metabolic homeostasis in IL-25–induced beige fat mice.** (A–C) Mice with DIO mice ($n$ = 5) administrated with clodronate-loaded liposomes to deplete macrophages and then injected with vehicle or IL-25 (1 μg/day) for 14 days. (A) Changes in fasting blood glucose. (B) GTT was conducted by intraperitoneal injection of glucose (2 g·kg$^{-1}$) and measurement of blood glucose concentration with a OneTouch Ultra Glucometer at designed time points in overnight-fasted mice. (C) ITT was done by IP injection of insulin (0.75 U·kg$^{-1}$) and measurement of blood glucose concentration by a OneTouch Ultra Glucometer at designed time points in 8 h–fasted mice. $^{*}p < 0.05$, compared with HFD-control-PBS group by two-sided unpaired $t$ test. Data present as mean ± SEM. (D–F) WT (UCP1$^{+/+}$) and UCP1-null (UCP1$^{-/-}$) mice ($n$ = 5 per treatment) were fed with an HFD for 12 weeks and then injected with IL-25 (1 μg) or vehicle for 14 days. (D) Changes in fasting blood glucose in these mice. (E) GTT was conducted by intraperitoneal injection of glucose (2 g·kg$^{-1}$) and measurement of blood glucose concentration with a OneTouch Ultra Glucometer at designed time points in overnight-fasted mice. (F) ITT was done by IP injection of insulin (0.75 U·kg$^{-1}$) and measurement of blood glucose concentration by a OneTouch Ultra Glucometer at designed time points in 8 h–fasted mice. $^{*}$# $p < 0.05$, compared with WT-HFD-IL-25 group, $^{*}p < 0.05$, compared with WT-HFD-PBS group by two-sided unpaired $t$ test. Data present as mean ± SEM. The data underlying this figure can be found in S1 Data. AUC, area under the curve; DIO, diet-induced obesity; GTT, glucose tolerance test; HFD, high-fat diet; IL, interleukin; IP, intraperitoneal; ITT, insulin tolerance test; NS, not significant; UCP1, uncoupling protein 1; WT, wild-type.

DIO that were injected with clodronate-loaded liposomes, suggesting that the antidiabetic effect of IL-25 was dependent on macrophages. However, IL-25 injection could reduce body weight gain (S3C Fig) and eWAT mass (S3D Fig), lower liver weight (S3E Fig) in mice with DIO that were injected with clodronate-loaded liposomes, suggesting that macrophages are not necessary for all of the anti-obesity effects activated by IL-25.

To investigate whether IL-25 protected mice with DIO from obesity and metabolic associated disorder by stimulating the development of beige fat, WT (UCP1$^{+/+}$) and UCP1-null (UCP1$^{-/-}$) mice were fed with an HFD for 12 weeks and then injected with IL-25 (1 μg/mouse) or vehicle for 14 days. From the glucose tolerance test (GTT) (S3F Fig) and insulin tolerance test (ITT) (S3G Fig), we could see the absence of UCP1, which made insulin sensitivity worse. The IL-25–mediated improvement in glucose clearance was abrogated in obese UCP1$^{-/-}$ mice, as IL-25 did not reduce fasting blood glucose (Fig 7D) or improve glucose tolerance (Fig 7E) and insulin sensitivity (Fig 7F) in obese UCP1$^{-/-}$ mice. No detectable difference in body mass was found between UCP1$^{+/+}$ and UCP1$^{-/-}$ mice fed with an HFD (S3H Fig). Genetic ablation of UCP1 may not influence the IL-25–mediated anti-obesity effect because IL-25 reduced body weight gain (S3I Fig) and eWAT mass (S3J Fig), lowered liver mass (S3K Fig) in obese UCP1$^{+/+}$ as effectively as in obese UCP1$^{-/-}$ mice. These results indicated that UCP1 played an important role in IL-25–mediated antidiabetic effect but not in effects on lowering body weight.

## Discussion

In this study, we identified that IL-25 signaling including IL-25 and its receptor IL-17RB increased in mice that developed beige adipose tissue induced by a β3-adrenoceptor agonist and cold exposure. Administration of IL-25 promoted the browning of WAT associated with significantly less weight gain and improved glucose and insulin tolerance in HFD-fed obese mice. Importantly, IL-25 may induce beige fat and enhance adipose tissue thermogenesis through the action of alternatively activating macrophages (polarizing from M1 to M2) to

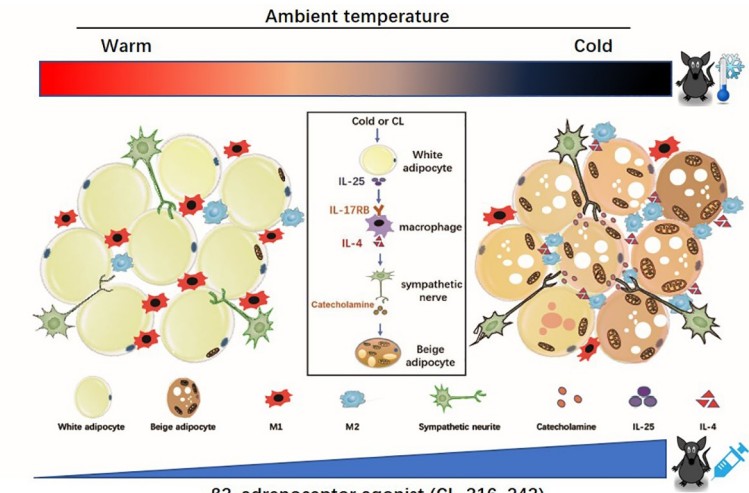

**Fig 8. Working model.** Cold stimulation or β3-adrenoceptor agonist (CL) treatment increased the expression of IL-25 signaling (IL-25 and its receptor IL-17RB) in the WAT. IL-25 can induce the beige *trans*-differentiation of white adipocyte and enhance adipose tissue thermogenesis by alternatively activating macrophages (polarizing from M1 to M2) to increase the level of catecholamine or release IL-4 exclusive of the simple and direct effect of IL-25 on adipocytes. CL, CL-316, 243; IL, interleukin; IL-17RB, IL-17 receptor B; WAT, white adipose tissue.

increase the level of catecholamine exclusive of the simple and direct effect of IL-25 on adipocytes (Fig 8). These data provided evidence that alternatively activated macrophages were involved in the development of beige fat, and IL-25, serving as a cytokine that connects alternatively activated macrophages and thermogenesis in mice, might play a potential therapeutic role in obesity and associated metabolic disorders.

Our study showed that both β3-agonist (1 mg/kg body weight for 2 days) and cold exposure (4°C for 2 days) led to a robust browning phenotype as well as activating IL-25 signaling in scWAT, whereas in eWAT, only β3-agonist led to the beige effect and cold exposure did not. Jia and colleagues demonstrated different cold-induced changes of UCP1 expression in BAT and WAT depots, which supported the notion that the effects of short-term cold exposure increased thermogenic capacity of BAT, as well as browning of scWAT and, to a limited extent, in eWAT [28]. Similarly, IL-25–induced beige fat formation appears initially in scWAT such as inguinal WAT (iWAT). Because cold exposure failed to induce beige fat in eWAT in <2 days and thermogenesis differs between scWAT and eWAT, differential development of beige fat in discrete BAT and WAT depots may be related to different response to IL-25. Although IL-25 signaling participated in the development of beige fat from WAT that included sharing the biochemical (mitochondrial biogenesis and high UCP1) and morphological (robust OXPHOS immunostaining and multilocular lipid droplets) characters with BAT [29], BAT UCP1 level were unchanged, suggesting that IL-25 could not affect the function of BAT. In addition to stimulating thermogenesis, IL-25 suppressed the expression of pro-inflammatory cytokines. Previous studies had demonstrated that the browning of WAT was closely related with changes in the expression of inflammatory genes [30,31] and that obese mice displayed higher expression of pro-inflammatory genes [32].

Our study comprehensively explored the mechanism involved in the process of IL-25–induced development of beige fat. IL-25 is considered to be different from other peptide-like cytokine inducers, such as irisin that directly acts on adipocytes to stimulate the development of WAT [33]. For example, we directly applied IL-25 on isolated APs in vitro, but differentiation of APs to beige adipocytes was not induced. When APs were co-cultured with peritoneal macrophages, however, IL-25 promoted APs differentiation toward beige adipocytes in vitro. Importantly, this phenomenon was blocked when anti-IL-4Rα neutralizing antibody was added to the co-culture medium, suggesting that IL-25 induced APs development to beige dependent on IL-4 and IL-4Rα. IL-4Rα is an essential receptor for IL-4/IL-13 induction of alternatively activated macrophages [34], and IL-4 can activate the IL-4Rα signaling in APs and promote the APs to develop the beige fat lineage [35]. We validated the fact that IL-25 induced alternatively activated macrophages to release IL-4 in vitro that activated the IL-4Rα signaling in APs and promoted development to the beige fat lineage. Notably, blockade of IL-4Rα in vivo significantly blunted IL-25–induced expression of UCP1. Collectively, these data suggested that IL-25 regulated the browning of WAT via alternatively activated macrophages and the effects of IL-4 signaling pathway on APs.

We found that blockade of IL-4Rα could not entirely inhibit IL-25–induced expression of UCP1, suggesting that IL-25 might participate in the development of beige fat through other mechanism. Prominent regional variation in beige fat biogenesis in scWAT with localization of UCP1+ beige adipocytes to areas with dense sympathetic neurites, and the density of sympathetic projections dependent on PRDM16 in adipocytes, provided another potential mechanism of development of beige fat mediated by PRDM16 and UCP1 [36]. The development of beige fat requires noradrenergic stimulation from sympathetic nerve system [37]. NE production by axons that express TH have been shown to be important for this process [38]. Hypothalamus sensing of cold exposure results in enhanced sympathetic nerve branch activation that induces the browning of WAT [38]. TH is mainly present in the cytosol and, to some

extent, in the neuron plasma membrane and catalyzes the rate limiting step in the initial reaction for the biosynthesis of catecholamines such as NE that are packaged in vesicles and exported through the synaptic membrane. NE released through innervation has been shown to be important for the browning of WAT. We documented that IL-25 enhanced the sympathetic innervation and then induce the expression of NE in WAT. It was reported that alternatively activated macrophages synthesize NE to sustain adaptive thermogenesis [39]. Whether alternatively activated macrophages are involved in the production of catecholamines to sustain adaptive thermogenesis seems controversial because they were reported to not synthesize catecholamines or contribute to adipose tissue adaptive thermogenesis [40]. We directly applied IL-25 on peritoneal macrophages in vitro to induce alternatively activated macrophages but failed to induce the expression of TH. This result suggested that IL-25 could not produce NE through alternative activation of macrophages. It was recently illustrated that BAT macrophages controlled tissue innervation [41]. In this study, we administered IL-25 to mice with DIO and injected with clodronate-loaded liposomes to deplete macrophages and found that impaired IL-25–induced sympathetic nerve branching and browning of WAT. Therefore, it appeared that IL-25 induced macrophages to regulate WAT innervation and the release of catecholamines such as NE for the development of beige fat. It appears that cytokine modulation of the immune environment in adipose tissue can stimulate the biogenesis of beige fat. This suggests that the accumulation and activation of type 2 innate lymphoid cells (ILC2s) by IL-33 could induce the browning of WAT [35,42].

IL-25 requires UCP1 to promote glucose clearance to maintain homeostasis. The effect of enhanced IL-25–mediated improvement in glucose clearance showed a marked dependence on macrophages, while the effect of IL-25 on body weight was not completely dependent on macrophages and beige fat. This may be related to the presence of other IL-25–responsive cells in the body, such as ILC-2 cells and eosinophils [43]. The activation of ILC-2 cells by IL-33 also induces the biogenesis of beige fat [35,42], and IL-25 can promote lipolysis by directly up-regulating the expression of some lipid metabolism-related enzymes, such as lipolytic enzymes (ATGL, MAGL, p-HSL) increased and lipogenic enzymes (ACC) reduced [26]. Although IL-25 could induce the browning effect, it was noted that lipolysis in brown adipocytes was not essential for cold-induced thermogenesis in mice [44].

In conclusion, our results demonstrated that IL-25 induced beige fat formation via alternatively activated macrophage and decreased glucose disposal and insulin resistance. Our study indicated that the modulation of IL-25 signaling might play a potential therapeutic role against obesity and its associated metabolic disorders.

## Materials and methods

### Ethical approval

All animal studies were conducted with the approval of the Institutional Animal Care and Use Committee (IACUC) of Sun Yat-sen University (approval number: SCXK2011-0029) meted with the China guideline of GB/T 35892–2018.

### Animals and in vivo experiment

WT C57/BL6J mice were purchased from the center of laboratory animal of Sun Yat-sen University. UCP1$^{-/-}$ mice and IL-25$^{-/-}$ mice with C57/BL6J genetic background were purchased from the model animal research center of Nanjing University. All mice were maintained under 12-hour light–dark cycles with a designed environmental temperature (21˚C ± 1˚C). Four-week-old male mice were fed with NCD or HFD (60% kcal fat diet) for 12 weeks to facilitate obesity. Except HFD-fed mice, 8-week-old male mice were used in all experiments. For

cold exposure experiments, C57/BL6J mice were kept under controlled temperature (22˚C) at first and then moved to 4˚C in individual cages for 48 hours. For β3-adrenoceptor agonist treatment, C57/BL6J mice were IP injected with CL (1 mg·kg$^{-1}$) or vehicle once daily for indicated time points. For IL-25–inducing browning experiments in NCD- or HFD-fed mice, various dose of IL-25 or vehicle were injected IP once daily for 7 days or 14 days. For irisin-inducing browning experiments in NCD-fed mice, irisin (1 μg) or vehicle were injected IP once daily for 2 or 5 days. For anti-IL-4Rα neutralization antibody in vivo, 125 μg anti-IL-4Rα antibody or isotype control Ab with or without 1 μg IL-25 were diluted with PBS to volume of 0.3 ml and injected IP twice a week at day 1 and day 4. To deplete macrophages, clodronate-loaded liposomes or empty liposomes (0.2 ml/mouse) were injected IP once every 2 days starting 5 days before injection with IL-25 (1 μg/day). After IL-25–inducing browning in HFD-fed mice, glucose (2 g·kg$^{-1}$) or insulin (0.75 U·kg$^{-1}$) were injected IP to perform intraperitoneal glucose or ITTs in overnight-fasted or 8 h–fasted mice. After injection, blood glucose concentration was measured using a Glucometer at designed time points. For insulin signaling experiments, IL-25–injected HFD-fed mice were injected with insulin (0.5 U·kg$^{-1}$) through the inferior vena cava, and then iWAT, namely scWAT, eWAT, interscapular BAT, liver, and muscle were harvested within 10 minutes. Cohorts of ≥4 mice per genotype or treatment were assembled for all in vivo studies. All in vivo studies were repeated 2 to 3 independent times.

## Metabolic rate and physical activity

Oxygen consumption and physical activity were determined for WT and IL-25$^{-/-}$ mice at 12 weeks of age using CLAMS according to the manufacturer's instructions. The animals were acclimated to the system for 20 to 24 hours, and measurement of VO$_2$ and VCO$_2$ was performed during the next 24 hours. The mice were maintained at 24˚C under a 12-hour light/dark cycle. Food and water were available ad libitum. Voluntary activity was derived from the x-axis beam breaks monitored every 15 minutes. Heat production and the RER were calculated as described previously.

## Stromal vascular fraction (SVF) isolation

SVF from scWAT of C57/BL6J female mice at age 4 weeks old. scWAT were washed with PBS, minced, and digested with 0.1% type II collagenase in DMEM containing 3% BSA and 25 μg/ml DNase I for 30 minutes at 37˚C. During the digestion, the mixed solution was shaken by hand every 5 minutes. The mixed solution was filtered through 70 μm cell strainer and then centrifuged at 500$g$ for 5 minutes at 4˚C. The floating adipocytes were removed, and the pellets containing the SVF were resuspended in red blood cell lysis buffer for 5 minutes at 37˚C. Cells were centrifuged at 500$g$ for 10 minutes at 4˚C, and the pellets were resuspended in DMEM medium containing 10% FBS and penicillin/streptomycin (100 units/ml).

## Cell culture

3T3-L1 MBX cells were purchased from the ATCC. 3T3-L1 MBX cells were cultured and grown to confluence in DMEM supplemented with 10% FBS, penicillin/streptomycin (100 units/mL). Adipocytes (3T3-L1 MBX and SVF) differentiation were induced by the beige adipogenic MDI mixture in 10% FBS DMEM medium containing 5 μg·ml$^{-1}$ insulin, 0.5 mM isobutylmethylxanthine, 0.5 mM dexamethasone, 1 nM tri-iodothyronine (T$_3$), 125 μM indomethacin, and 1 μM rosiglitazone. Two days after induction, the media was switch to the maintenance medium containing 10% FBS, 1 nM T$_3$, and 5 μg·ml$^{-1}$ insulin for another 6 days.

Various doses of IL-25 and CL (10 μM) were added when cells reached confluence and sustained through 8 days.

## Macrophage isolation

Macrophages were induced into the peritoneal cavity by injection of 100% mineral oil (0.5 ml). After washing with PBS, macrophages were cultured overnight in DMEM with 10% FBS.

## Real-time PCR

Total RNA from tissue or cells was extracted with RNA isolation reagent. RNA concentration was measured by spectrometer. Quantity of 1,000 ng total RNA was reverse transcribed into cDNA by RT reagent Kit Perfect Real Time kit. RNA-time PCR analysis using SYBR-Green fluorescent dye was performed with a real-time fluorescence quantitative PCR instrument.

## Histology and immunohistochemistry

eWAT, scWAT, and interscapular BAT were fixed in 4% paraformaldehyde. Tissues were embedded with paraffin and sectioned by microtome. The slides were stained with hematoxylin and eosin (HE) using standard protocol. For UCP1 immunohistochemistry, slides of various tissue were blocked with goat serum for 1 hour. Subsequently, the slides were incubated with anti-UCP1 (1:1,000; ab10983) overnight at 4˚C followed by detection with the DAB Horseradish Peroxidase Color Development Kit. Hematoxylin was used as counterstain.

## Immunofluorescence

For immunofluorescence staining, slides were incubated with rabbit anti-mouse IL-17RB (1:1,000; H-40), rabbit anti-mouse UCP1(1:1,000; ab10983), rat anti-mouse IL-25 (1:1,000; MAB1399), NF (1:1,000; ab8135), and TH (1:1,000; ab112) overnight at 4˚C, followed by staining with a mixture of secondary antibodies containing an Alex Flour 488-Donkey anti-rat IgG (H+L) (1:200; A21208) and an Alex Flour 594-Donkey anti-rabbit IgG (1:200; R37119) for 1 hour at room temperature. The cell nuclei were counterstained with 4′, 6-diamidino-2-phenylindole (DAPI) for 15 minutes at room temperature. The slides were observed with a confocal laser scanning microscope.

## Immunoblot analysis

Tissues and cells were lysed in RIPA buffer supplemented with 1 mM PMSF. The protein concentration was measured by the BCA protein assay kit, and total cellular protein (30 μg) were subject to western blot analysis. The protein was transferred to PVDF membrane and incubated with primary antibodies against HSP90 (1:1,000; C45G5), UCP1(1:1,000 for eWAT, scWAT, and cells, or 1:10,000 for BAT; ab10983) (The auto exposure time of eWAT is significantly longer than scWAT), TH (1:1,000; ab112), IL-17RB (1:1,000; H-40), p-AKT (1:1,000; 4060S), and AKT (1:1,000; 4691S). After incubated with goat anti-rabbit IgG/HRP (1:1,000; PI1000) secondary antibody, proteins were detected with chemoluminescence using Immobilon Western HRP Substrate.

## ELISA

Catecholamine level was detected using a sensitive ELISA kit (CSB-E07870m). The IL-25 level of mouse was detected by Mouse IL-25 ELISA kit (DY1399). Triglycerides (TG) and free fatty acids (FFAs) in plasma were measured by TG assay kit (ETGA-200) and FFA assay kit (EFFA-100). The plasma insulin level of mice was detected using a sensitive ELISA kit (10-1247-01).

All measurements were performed with standard manufacture protocol. Adipose tissue (100 mg) was rinsed and homogenized in 1 ml of PBS. The homogenates were centrifuged for 5 minutes at 10,000 rpm for 10 minutes at 4˚C, and then the supernatant was assayed immediately. All samples were normalized to total tissue protein concentration.

## Statistical analysis

All data are presented as mean ± SEM. Student $t$ test was used to compare between 2 groups, and one-way ANOVA followed by LSD-$t$ test was applied to compare more than 2 different groups on GraphPad Prism software. $p < 0.05$ was considered significant.

## Supporting information

**S1 Fig. IL-25 signaling increased in epididymal beige adipose tissue induced by β3-adrenergic agonist stimulation or cold exposure.** (A–F) WT mice were injected with CL (1 mg/kg body weight) for 2 (CL-2d) and 5 days (CL-5d). (A) The protein level of UCP1 was analyzed by western blot in WAT. HSP90 was used as a loading control. (B) H&E staining of eWAT. 400× magnification. (C) Immunohistochemical staining for UCP1 of eWAT. 400× magnification. (D) The protein level of IL-17RB was analyzed by western blot in eWAT. HSP90 was used as a loading control. (E) Immunofluorescent staining for IL-25 (IL-25$^+$ green) and UCP1 (UCP1$^+$ red) or IL-17RB (IL-17RB$^+$ red) of eWAT. Nucleus stained with DAPI (blue). Images were photographed at 200× magnification. (F) IL-25 protein expression in eWAT ($n$ = 4–5 per treatment). (G–L) WT mice were placed at 22˚C or 4˚C in individual cages for 48 hours ($n$ = 4–5 per treatment). (G) The protein level of UCP1 was analyzed by western blot in WAT. HSP90 was used as a loading control. (H) H&E staining of eWAT in 22˚C or 4˚C for 48 hours. 400× magnification. (I) Immunohistochemical staining for UCP1 of eWAT at 22˚C or 4˚C for 48 hours. 400× magnification. (J) The protein level of IL-17RB was analyzed by western blot of eWAT at 22˚C or 4˚C for 48 hours. HSP90 was used as a loading control. (K) Immunofluorescent staining for IL-25 (IL-25$^+$ green) and UCP1 (UCP1$^+$ red) or IL-17RB (IL-17RB$^+$ red) of eWAT at 22˚C or 4˚C for 48 hours. Nucleus stained with DAPI (blue). Images were photographed at 200× magnification. (L) Results of RT-qPCR analysis showing the expression of IL-25 in eWAT and liver of mice maintained at 22˚C or 4˚C for 48 hours. ($n$ = 4–5). $^*p < 0.05$ by two-sided unpaired $t$ test. Data present as mean ± SEM. The data underlying this figure can be found in S1 Data. CL, CL-316, 243; eWAT, epididymal WAT; H&E, hematoxylin and eosin; IL, interleukin; IL-17RB, IL-17 receptor B; RT-qPCR, Reverse transcription - quantitative PCR; scWAT, subcutaneous WAT; UCP1, uncoupling protein 1; WAT, white adipose tissue; WT, wild-type.
(TIF)

**S2 Fig. IL-25 played an important role in the improvement of metabolic homeostasis during the cold exposure.** WT and IL-25$^{-/-}$ mice were placed at 22˚C or 4˚C for 48 hours. (A) Changes in body mass of mice. (B) The daily food intake. (C) The temperature of mice after treatment. (D) The TG level and (E) the FFA level of mice after treatment. (F) Changes in fasting blood. (G) Changes of fasting insulin level in mice. (H) GTT. (I) ITT was conducted by IP injection of glucose (1 g·kg$^{-1}$) or injection of insulin (1 U·kg$^{-1}$) and measurement of blood glucose concentration in overnight-fasted mice. The tissues were harvested within 10 minutes after an injection of insulin (1 U·kg$^{-1}$). $^*p < 0.05$, compared with control group by two-sided unpaired $t$ test. Data present as mean ± SEM. The data underlying this figure can be found in S1 Data. FFA, free fatty acid; acid; GTT, glucose tolerance test; IL, interleukin; IP,

intraperitoneal; ITT, insulin tolerance test; NS, not significant; TG, triglyceride; WT, wild-type.
(TIF)

**S3 Fig. Depletion of macrophages and genetic ablation of UCP1 did not block IL-25–mediated lowering of body weight.** (A) The western blot analysis level of UCP1 protein in scWAT and eWAT from mice injected with vehicle and various doses of IL-25 for 14 days. (B) Depletion of macrophage with clodronate and the mRNA expression of F4/80 examined by RT-qPCR in scWAT. (C–E) DIO mice ($n = 5$) administered with clodronate-loaded liposomes to obliterate macrophages and then injected with vehicle or IL-25 (1 μg/day) for 14 days. Changes in body mass (C), eWAT mass (D), and liver mass (E) from different treatment groups. (F–K) WT (UCP1$^{+/+}$) and UCP1-null (UCP1$^{-/-}$) mice ($n = 5$ per treatment) were fed with an HFD for 12 weeks and then injected with IL-25 (1 μg) or vehicle for 14 days. (F) GTT was conducted by oral glucose (2 g·kg$^{-1}$) and measurement of blood glucose concentration with OneTouch Ultra Glucometer at designed time points in overnight-fasted mice. (G) ITT was done by IP injection of insulin (0.75 U·kg$^{-1}$) and measurement of blood glucose concentration by One-Touch Ultra Glucometer at designed time points in 8 h–fasted mice. (H) Weight of WT (UCP1$^{+/+}$) and UCP1-null (UCP1$^{-/-}$) mice assessed after feeding HFD. (I) Changes in body mass. (J) eWAT mass. (K) Liver mass from different treatment groups. #$p < 0.05$, compared with UCP1$^{-/-}$-HFD-PBS group, *$p < 0.05$, compared with WT-HFD-PBS group by two-sided unpaired $t$ test. Data present as mean ± SEM. The data underlying this figure can be found in S1 Data. DIO, diet-induced obesity; eWAT, epididymal WAT; GTT, glucose tolerance test; HFD, high-fat diet; IL, interleukin; IP, intraperitoneal; ITT, insulin tolerance test; RT-qPCR, quantitative real-time PCR; scWAT, subcutaneous WAT; UCP1, uncoupling protein 1; WT, wild-type.
(TIF)

**S1 Data. Data underlying Figs 1–8 and S1–S3 Figs.**
(XLSX)

**S1 Raw Images. Uncropped blots underlying Figs 1B, 1E, 1K, 2A, 2D, 2I, 3E, 3F, 3H, 4A, 4B, 4G, and 5H and S1A, S1D, S1G, S1J, and S3A Figs.**
(PDF)

## Author Contributions

**Conceptualization:** Lingyi Li, Lei Ma, Zewei Zhao, Weiwei Qi, Xia Yang, Guoquan Gao, Zhonghan Yang.

**Data curation:** Lingyi Li, Lei Ma, Zewei Zhao, Shiya Luo, Baoyong Gong, Jin Li, Juan Feng, Hui Zhang, Weiwei Qi, Ti Zhou, Xia Yang, Zhonghan Yang.

**Formal analysis:** Lei Ma, Zewei Zhao, Shiya Luo, Baoyong Gong, Jin Li, Juan Feng, Hui Zhang, Weiwei Qi, Ti Zhou, Xia Yang, Zhonghan Yang.

**Funding acquisition:** Weiwei Qi, Ti Zhou, Xia Yang, Guoquan Gao, Zhonghan Yang.

**Investigation:** Guoquan Gao, Zhonghan Yang.

**Methodology:** Lingyi Li, Zewei Zhao, Shiya Luo, Baoyong Gong, Juan Feng, Hui Zhang.

**Resources:** Lingyi Li, Zhonghan Yang.

**Supervision:** Guoquan Gao, Zhonghan Yang.

Writing – original draft: Lingyi Li, Lei Ma.

Writing – review & editing: Zhonghan Yang.

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
