## [Editor Report · Decision Letter 0]

11 Sep 2020

Dear Dr Yang, 

Thank you for submitting your manuscript entitled "IL-25 induces beige fat to improve metabolic homeostasis via macrophage and innervation" for consideration as a Research Article by PLOS Biology.

Your manuscript has now been evaluated by the PLOS Biology editorial staff and I am writing to let you know that we would like to send your submission out for external peer review.

Please re-submit your manuscript within two working days, i.e. by Sep 15 2020 11:59PM.

Kind regards,

Lucas Smith, Ph.D.,

Associate Editor

PLOS Biology

---

## [Decision Letter · Decision Letter 1]

25 Nov 2020

Dear Dr Yang,

Thank you very much for submitting a revised version of your manuscript " IL-25 induces beige fat to improve metabolic homeostasis via macrophage and innervation" for consideration as a Research Article at PLOS Biology. This revised version of your manuscript has been evaluated by the PLOS Biology editors, the Academic Editor and the original reviewers.

The reviews of your manuscript are appended below. You will see that while reviewer 2 feels that your revised manuscript has addressed most of his/her initial concerns and is largely satisfied, reviewers 1 and 3 still have lingering concerns, which would need to be addressed before we could consider this manuscript for publication. For example, reviewer 3 notes that some of the data raises questions about the beneficial role of IL-25 in metabolic disorders. Reviewer 1 has raised concerns regarding a potential contribution of autoflourescent signals, which could affect the conclusions of the study, and notes that the conclusion that IL-25 regulates TH innervation is not fully supported.

In light of the reviews, we will not be able to accept the current version of the manuscript, but we would welcome re-submission of a much-revised version that takes into account the reviewers' comments. Also, we would like you to add the data that you included in your response to the reviewers' comments in the revised manuscript as supplementary figures (for example include the figures presented in response reviewer 3's point 5, 6, and 9).We cannot make any decision about publication until we have seen the revised manuscript and your response to the reviewers' comments. Your revised manuscript is also likely to be sent for further evaluation by the reviewers.

We expect to receive your revised manuscript within 3 months. 

**IMPORTANT - SUBMITTING YOUR REVISION**

*Re-submission Checklist*

*Published Peer Review*

*PLOS Data Policy*

*Blot and Gel Data Policy*

Sincerely,

Lucas Smith, Ph.D.,

Associate Editor,

lsmith@plos.org,

PLOS Biology

REVIEWS:

Reviewer's Responses to Questions

PLOS authors have the option to publish the peer review history of their article (what does this mean?). If published, this will include your full peer review and any attached files.

Reviewer #1: No

Reviewer #2: No

Reviewer #3: No

Reviewer #1: The authors conducted a substantial amount of additional experiments and addressed most, if not all, of the questions and concerns that were raised. The revised manuscript has been significantly improved for publication with more careful language editing.

Reviewer #2: The authors have not satisfactorily addressed some major questions. For examples, The IL-25-injected animals were more glucose tolerant and insulin sensitive. You would expect to see reduced blood insulin levels. Unfortunately, the opposite was observed according to the rebuttal, which raises a serious question about the beneficial role of IL-25 in metabolic disorders. It is highly possible that IL-25-injected animals had some issues in the gastrointestinal organ causing less weight gain, which affected some metabolic parameters. IL-25 KO mice only showed modest changes in weight gain and ITT. Also it is unclear why the high fat-fed control mice weighed less than 25 grams. The beiging effect (Fig. 6B) is quite unusual because 48h cold exposure cannot induce that much beiging morphologically.

Reviewer #3: In the current study, Li et al. reported that IL-25 could regulate the white adipose tissue (WAT) browning process. IL25 and IL-17RB protein levels increased by the stimulation of b3-adrenergic receptor agonist or cold exposure. Administration of recombinant IL-25 could induce white adipose tissue browning and alternatively activated macrophages were proposed to be IL-25 responsive cells during this process. The authors further showed IL-25 treatment improved glucose metabolism in the high-fat diet-fed animals, and IL-25 knockout mice were protected from DIO. Overall, the data presented here indicate an interesting role that IL-25 plays to regulate WAT browning. 

Specific comments: 

1. The description should be more accurate and inclusive in the introduction. For examples, "although the thermogenic tissue is almost lacking in adults" should refer to human and adaptive thermogenesis of classical brown adipose tissues, and regarding the lineage of beige adipocytes, the Myf5 positive cells have been shown to contribute partly to brite/beige cells in the scWAT. The discussion should be more coherent and focused. 

2. The role of Ucp1 vs beige fats was not clearly explained. The improvement of glucose metabolism in DIO by IL-25 administration was dependent on Ucp1 and macrophages as in Fig7. But the body weight change and the fat pad decrease induced by IL-25 seems largely independent of Ucp1 or macrophages as in FigS3. 

3. The adipose tissues normally show high auto-fluorescent signals. To rule out the contribution of autofluorescence from adipocytes in immunostaining of IL-25, IL-17RB, and CD68, high-resolution images, and negative controls should be included. And isolated adipocytes and adipose macrophages using FACS sorting under cold challenge or CL stimulation would provide both gene expression and western blot data to consolidate the conclusion from immunostaining. 

4. The conclusion of IL-25 on the regulation on TH innervation is not founded by the current dataset. It could be a correlation between browning and TH innervation change. TH staining in Fig. 4D and 4F showed different morphology. The discrepancy in the results needs to be clarified. 

5. Lack of scale bar in Fig. 1I; 1J; Fig. 4F; Fig. 6B.

6. Typo: mRNA expression of scWAT (arbitrary units) from Fig. 2B to Fig. S3B.

7. S1F, the figure should show how the data were normalized. 

8. The description of UCP1 expression levels in eWAT and western blots in various figures was not consistent.

---

## [Decision Letter · Decision Letter 2]

4 Jun 2021

Dear Dr Yang,

Thank you for submitting your revised Research Article entitled "IL-25 induces beige fat to improve metabolic homeostasis via macrophage and innervation" for publication in PLOS Biology. I have now obtained advice from two of the original reviewers and from the Academic Editor. 

As you will see, the reviewers are largely satisfied by your revision. Therefore, we will probably accept this manuscript for publication, provided you satisfactorily address the remaining points raised by the reviewers. **IMPORTANT: Please also make sure to address the following data and other policy-related requests.

1) Please address reviewer 3's comment that some words within the figures are distorted.

2) As this will be the last chance to do so, please carefully read and edit the paper for grammar and clarity. It may be helpful to run your manuscript by a native English speaker, in order to ensure accessibility to a broad readership.

3) Having discussed the title of your manuscript with my colleagues, we wonder if it might be edited slightly to improve clarity. If you agree, we might suggest something like "IL-25-induced shifts in macrophage polarization promote fat beigeing and improve metabolic homeostasis in mice".

4) FINANCIAL DISCLOSURES: Where possible, please provide the URLs of the websites for each funder listed in your financial disclosures statement. 

5) ETHICS REQUESTS: Please include the specific national or international regulations/guidelines to which your animal care and use protocol adhered. Please note that institutional or accreditation organization guidelines (such as AAALAC) do not meet this requirement.

6) ETHICS REQUEST: Please provide the identification number of the protocol, approved by the Ethics Committee of the Guangzhou Women and Children’s Hospital and the First Affiliated Hospital of Sun Yat-sen University.

7) BLURB: Please also provide a blurb which (if accepted) will be included in our weekly and monthly Electronic Table of Contents, sent out to readers of PLOS Biology, and may be used to promote your article in social media. The blurb should be about 30-40 words long and is subject to editorial changes. It should, without exaggeration, entice people to read your manuscript. It should not be redundant with the title and should not contain acronyms or abbreviations. For examples, view our author guidelines: https://journals.plos.org/plosbiology/s/revising-your-manuscript#loc-blurb

8) DATA AVAILABILITY REQUEST: Please provide, as a Deposition in a publicly available repository, the data underlying each figure (including supplemental figures). Please be sure to reference this file in each figure legend. For example, to each figure legend you could add a statement saying "The data underlying this figure can be found in S1_Data. Please also ensure this file has a legend describing what it is. Find more information on this request below my signature.

9) DATA AVAILABILITY REQUEST: Please provide the original, uncropped and minimally adjusted images supporting all blot and gel results reported in an article's figures or Supporting Information files. Please ensure these comply with our guidelines for how to prepare and upload this data: https://journals.plos.org/plosbiology/s/figures#loc-blot-and-gel-reporting-requirements.Find more information on this request below my signature. 

We expect to receive your revised manuscript within two weeks. 

*Published Peer Review History*

*Early Version*

Sincerely,

Lucas Smith, Ph.D.,

Associate Editor,

lsmith@plos.org,

PLOS Biology

DATA POLICY:

Figure 1 G,H,L; Figure 2 B,C,G,H; Figure 3 A-D,G; Figure 4C,E; Figure 5A,D,E-G; Figure 6 A,C-I; Figure 7A-F

Figure S1 F,L; Figure S2 A-I; Figure S3 B-K

We require the original, uncropped and minimally adjusted images supporting all blot and gel results reported in an article's figures or Supporting Information files. We will require these files before a manuscript can be accepted so please prepare and upload them now. Please carefully read our guidelines for how to prepare and upload this data: https://journals.plos.org/plosbiology/s/figures#loc-blot-and-gel-reporting-requirements 

Reviewer remarks:

Reviewer #2: Authors appropriately addressed my previous comments.

Reviewer #3: The questions are addressed. Some details: for example, some words within the figures are distorted.

---

## [Editor Report · Decision Letter 3]

2 Jul 2021

Dear Dr Yang,

On behalf of my colleagues and the Academic Editor, Ligong Chen, I am pleased to say that we can in principle offer to publish your Research Article "IL-25-induced shifts in macrophage polarization promote development of beige fat and improve metabolic homeostasis in mice" in PLOS Biology, provided you address any remaining formatting and reporting issues. These will be detailed in an email that will follow this letter and that you will usually receive within 2-3 business days, during which time no action is required from you. Please note that we will not be able to formally accept your manuscript and schedule it for publication until you have made the required changes.

Thank you very much for addressing our editorial requests in the last revision. I have uploaded the revised S1_data file and S1_raw images file that you provided to the manuscript. When looking through your manuscript, I did notice what I think is a typo on line 248 - Line 248 currently says "Macrophage depletion and UCP1 gene knockout (IL-25-/-) ameliorated IL-25- mediated improvement in glucose homeostasis in mice" - I think it should be "Macrophage depletion and UCP1 gene knockout (UCP1-/-) ameliorated IL-25- mediated improvement in glucose homeostasis in mice". If you agree, you can fix this typo while addressing the other formatting checks in the email to come. 

PRESS

Sincerely, 

Lucas Smith, Ph.D. 

Senior Editor 

PLOS Biology

lsmith@plos.org